# Magnetic Inversion through a Modified Adaptive Differential Evolution

**Tao Song [1], Lianzheng Cheng [2,3,*] , Tiaojie Xiao [4], Junhao Hu [5,*] and Beibei Zhang [5]**

1    College of Computer Science, Guiyang University, Guiyang 550005, China; songt1988@outlook.com
2    School of Earth Sciences, Yunnan University, Kunming 650500, China
3    Yunnan Key Laboratory of Statistical Modelling and Data Analysis, School of Mathematics and Statistics, Yunnan University, Kunming 650091, China
4    Science and Technology on Parallel and Distributed Processing Laboratory, National University of Defense Technology, Changsha 410073, China; xiaotiaojie@nudt.edu.cn
5    College of Architectural Science and Engineering, Guiyang University, Guiyang 550005, China; zhangbeibei28@126.com
*    Correspondence: chenglianzh10@mails.ucas.ac.cn (L.C.); hujunhao2013@outlook.com (J.H.)

**Abstract:** In recent decades, differential evolution (DE) has been employed to address a diverse range of nonlinear, nondifferentiable, and nonconvex optimization problems. In this study, we introduce an enhanced adaptive differential evolution algorithm to address the inversion problem associated with magnetic data. The primary objective of the inversion process is to minimize the discrepancy between observed data and predicted data derived from the inverted model. So, the contributions of this paper include the following two points. First, a new mechanism for generating crossover rate (CR) is proposed, which tends to reduce the CR values corresponding to vectors with better objective function values. Second, a new mutation strategy with direction information is proposed to expedite convergence. Additionally, modifications were made to the adjustment of the regularization factor to prevent it from becoming too minimal, thereby preserving its efficacy. The proposed algorithm is validated through synthetic models and a field example. Results from synthetic models demonstrate that our method is superior to and competitive with the original adaptive DE in both solution quality and convergence velocity. For the field example, the Inverted models align closely with the drill-well information.

**Keywords:** differential evolution; magnetic inversion; regularization factor; inverse problem

## 1. Introduction

The magnetic method has been widely used in various fields such as petroleum and mineral resources. It is also employed in investigating the subsurface magnetic distribution within both the environmental and engineering fields [1]. In the realm of magnetic data processing, there are primarily two types of inversion methods, geometric parameter inversion [2] and physical property inversion [3–5]. Compared to geometric parameter inversion, physical property inversion necessitates the introduction of a regularization factor to reduce the number of potential models. This is due to the ill-posed nature of physical property inversion, where an infinite number of solutions can fit the observed data accurately [6]. In general, two types of optimization techniques can be employed to solve parametric inversion and physical property inversion: local optimization and global optimization. So far, both local optimization and global optimization have been used for magnetic inversion. For local optimization, in [7], a weighted–damped least-squares method was used for magnetic inversion. Zuo et al. [8] used the nonlinear conjugate gradient (NLCG) method to invert the amplitude of the magnetic anomaly vector. In global optimization, many metaheuristic algorithms, such as the genetic algorithm (GA) [9], particle swarm optimization (PSO) [10], simulated annealing (SA) [11], etc., have been widely

employed for inversion of magnetic data. For example, in literature [12,13], the magnetic data were inverted and interpreted by applying a PSO algorithm. Differential evolution, an evolution algorithm (EA), was designed by Storn and Price [14] for optimizing Chebyshev polynomial problems. The advantages of DE include simplicity, robustness, and speed. Hence, in recent decades, it has been extended to solve various problems from scientific and engineering domains, such as image reconstruction and denoising [15–19], optimal power flow [20–22], and neural network training [23,24], and it has also been applied in magnetic inversion [25–27]. Generally, an individual in the DE population is typically referred to as a target vector. These target vectors are continuously evolved through the repetitive application of mutation, crossover, and selection operations. Although DE benefits from its attractive advantages, there exist some shortcomings. For example, the control parameters, scale factor (*F*), and crossover rate (*CR*) are problem-dependent. Consequently, adapting them to each specific optimization problem presents a significant challenge.

Currently, JADE (an adaptive differential evolution) [28] has been proven to be a successful DE method and has been effectively applied to gravity and magnetic inversion [26,29]. However, from our perspective, there remains potential for JADE's performance to be further enhanced according to the studies by [26,29]. In this paper, we mainly focus on presenting an improved version of JADE, and we apply it to magnetic inversion. Firstly, in order to augment the convergence speed, a new mutation strategy with direction information is developed. In addition, as indicated in [30], by reassigning the crossover rate according to the fitness value of each target vector, we can maintain the gene information of top vectors. Inspired by this observation, a novel *CR* generation scheme on the basis of fitness value is used. Moreover, to adjust the role of data misfit and model misfit in the objective function of inversion, an appropriate regularization factor is critical. In previous studies [26,29], the regularization always decreases if the mean of data misfit increases in the DE population. It is plausible that the effect of model misfit will be eliminated if the regularization parameter is excessively small. Therefore, a threshold value is integrated into the regularization factor.

## 2. Theory and Method

### 2.1. Forward Magnetic Modelling

Set the discretized magnetic susceptibility of the underground model as vector **m**. Denote the forward modeling operator as $\mathcal{F}$. The magnetic data can then be acquired through the following:

$$\mathbf{d} = \mathcal{F}(\mathbf{m}), \tag{1}$$

where **m** indicates the magnetic susceptibility at each cell of a rectangular grid, and **d** represents the magnetic data. In the integral equation domain formulation, $\mathcal{F}$ is a rectangular matrix of size $N_d \times N_m$ dimensions [12,31,32], $N_d$ is the number of observation points, and $N_m$ is the mesh cells. However, in this study, $\mathcal{F}$ is nonlinear, as we use the finite volume method, which is a differential equation domain formulation [33].

### 2.2. Improved Differential Evolution

As stated in the Introduction, DE is a population-based metaheuristic EA which has been proved efficient for solving tough, nonlinear, and nondifferentiable optimization problems. In this part, the basic structure and our modifications of DE will be described.

#### 2.2.1. Classic Differential Evolution

Generally speaking, DE consists of four main steps: initialization, mutation, crossover, and selection, as shown in Figure 1; these can be found in previous papers in detail [25,27,29].

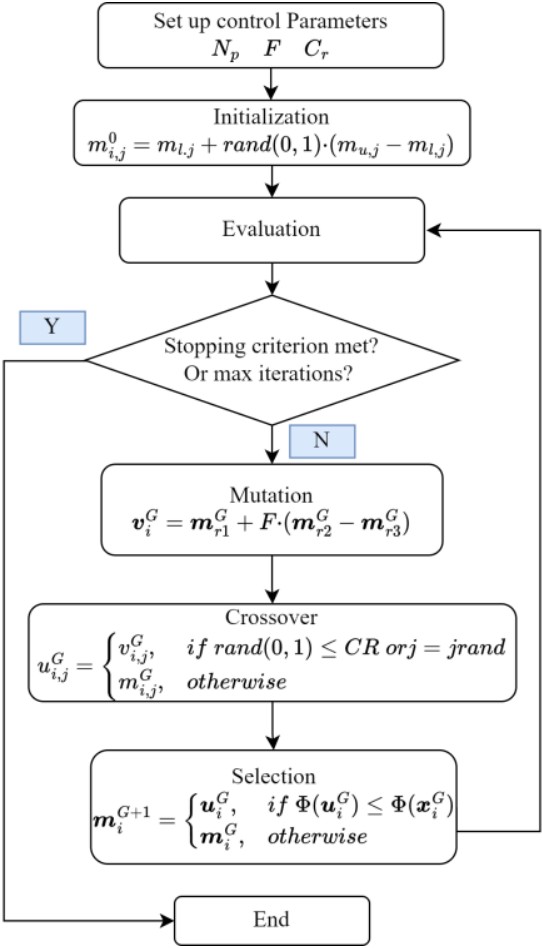

**Figure 1.** A simplified flowchart of the DE algorithm.

Generally, the optimization problem can be expressed as finding **m** so that the functional

$$\Phi(\mathbf{m}) = \|\mathbf{d} - \mathcal{F}(\mathbf{m})\|_2^2 \tag{2}$$

is minimized, where **d** represents the target data, and $\mathcal{F}(\mathbf{m})$ denotes the output when the input is **m**. The target vector with index $i$ ($i \in [1, NP]$) can be written as $\mathbf{m}_i = (m_{i,1}, m_{i,2}, \cdots, m_{i,M})$ in population **P**, where $M$ denotes the size of vector $\mathbf{m}_i$. After initialization, the vectors of DE are refreshed by mutation, crossover, and selection operation at each iteration. Currently, most of the improvements to the DE algorithm focus on developing new mutation strategies and designing novel adaptation mechanisms for control parameters that have led to the development of many significant algorithms [34,35]. The original mutation strategy can be expressed as follows:

$$\text{DE/rand/1}: \mathbf{v}_i^G = \mathbf{m}_{r1}^G + F \cdot \left( \mathbf{m}_{r2}^G - \mathbf{m}_{r3}^G \right), \tag{3}$$

Generally, the notation DE/x/y is used to denotes one variant of DE [14], where x specifies the vector to be mutated, and y is the number of difference vectors used, so DE/rand/1 in Equation (3) means a random population vector is chosen to be mutated, and one difference vector is used. In Equation (3) $r1$, $r2$, and $r3$ are mutually different integers uniformly generated from the range of $[1, NP]$, and all of them are not equal to $i$.

### 2.2.2. JADE Algorithm

Considering that our inversion work of magnetic data is based on a modified JADE [28] algorithm, a simplified flowchart of the JADE algorithm is shown in Figure 2. The novel-

ties of JADE include two aspects: the DE/current-to-pbest/1 mutation strategy [28] and adaptation of control parameters.

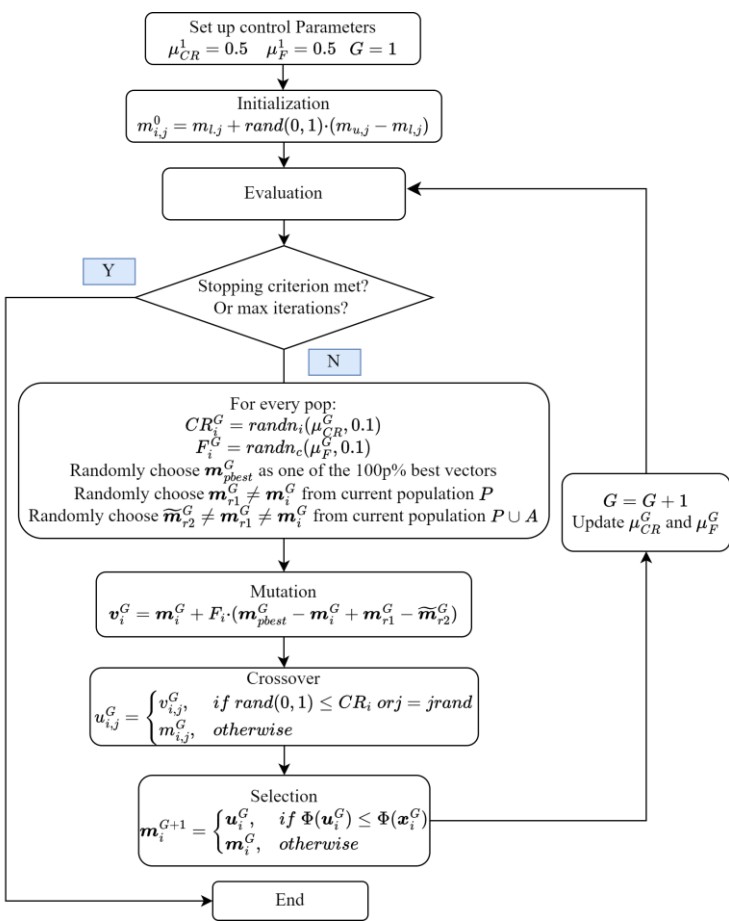

**Figure 2.** A simplified flowchart of the JADE algorithm.

- Mutation strategy.

    As shown in Figure 2, in the mutation stage, $\mathbf{m}_{pbest}^G$ is selected randomly among the top 100 $p$% vectors, and vector $\widetilde{\mathbf{m}}_{r2}^G$ is selected from the union of population **P** and archive **A**. The archive **A** is used to store the target vectors that failed in the selection operation.

- Control parameters.

    According to Figure 2, the parameters $F$ and $CR$ are generated by function $randc(\ )$ and $randn(\ )$, which represent the Cauchy distributions and the normal distributions, respectively. The value of $\mu_{CR}^G$ and $\mu_F^G$ is updated by using the following formula:

$$\begin{aligned} \mu_{CR}^G &= (1-c)\mu_{CR}^{G-1} + c\ mean_A(S_{CR}) \\ \mu_F^G &= (1-c)\mu_F^{G-1} + c\ mean_L(S_F) \end{aligned}, \tag{4}$$

where $c$ is a constant number between 0 and 1 (in the original JADE, $c = 0.1$), $mean_A$ is the arithmetic mean, and $mean_L$ denotes the Lehmer mean [28] and is described as follows:

$$mean_L(S_F) = \frac{\sum_{F \in S_F} F^2}{\sum_{F \in S_F} F} \tag{5}$$

where $S_{CR}$ and $S_F$ are the sets of all successful $CR$s and $F$s in the last iteration [28].

### 2.2.3. The Proposed JADE

As mentioned in the Introduction, the main modifications of our work on JADE include two aspects: a *CR* generation method and a mutation strategy with direction information.

- The new *CR* generation mechanism is as follows.

According to the statement of literature [36], a preferable direction for the search can be obtained through using high-quality vectors. Hence, the remaining good gene information in the top vectors will have the potential to improve the search ability of DE. Additionally, it is easier to generate a better solution if the number of components from mutant vectors is reduced. Based on these observations, the *CR* value of each individual is created according to the following formula:

$$
\begin{aligned}
CR_i^G &= \mu_{CR}^G + 0.1 \cdot \sigma\big(\mathbf{\Phi}, \Phi(\mathbf{m}_i^G)\big), \\
CR_i^G &= \begin{cases} 0.5 CR_i^{G-1} & \text{if } CR_i^G < 0 \\ 0.5\big(CR_i^{G-1} + 1\big) & \text{else if } CR_i^G > 1 \\ CR_i^G & \text{other wise} \end{cases}
\end{aligned}
\tag{6}
$$

where $\mathbf{\Phi}$ denotes a vector formed by all fitness values (i.e., $\mathbf{\Phi} = \big[\Phi(\mathbf{m}_1^G), \Phi(\mathbf{m}_2^G), \cdots, \Phi(\mathbf{m}_{NP}^G)\big]$). $\sigma(\cdot)$ returns the score value of $\mathbf{m}_i^G$, which is calculated by the following formula:

$$
\sigma\Big(\mathbf{\Phi}, \Phi\big(\mathbf{m}_i^G\big)\Big) = \frac{\Phi(\mathbf{m}_i^G) - \text{mean}_A(\mathbf{\Phi})}{\text{mean}_A(|\mathbf{\Phi} - \text{mean}_A(\mathbf{\Phi})|)},
\tag{7}
$$

From Equation (5), for a minimization problem, a small *CR* value will be assigned to the vector with better fitness. For example, based on the proposed scheme, the best vector will have the smallest *CR* value.

- Mutation strategy based on direction information is as follows.

In the original JADE, the search direction actually is calculated by two vectors selected randomly (i.e., $\mathbf{m}_{r1}^G$ and $\widetilde{\mathbf{m}}_{r2}^G$). Specifically, $\mathbf{m}_{r1}^G$ is sampled from population $\mathbf{P}$, and $\widetilde{\mathbf{m}}_{r2}^G$ is randomly chosen from the union population constructed by $\mathbf{P}$ and archive $\mathbf{A}$. In order to accelerate the convergence speed more effectively, a mutant vector is generated based on the following mutation formula:

$$
\begin{aligned}
v_i^G &= m_i^G + F_i \cdot \Big( m_{pbest}^G - m_i^G + \Delta_i \cdot \big( m_{r1}^G - \widetilde{m}_{r2}^G \big) \Big) \\
\Delta_i &= \begin{cases} 1 & \text{if } \Phi(m_{r1}^G) \leq \Phi\big(\widetilde{m}_{r2}^G\big) \\ -1 & \text{otherwise} \end{cases}
\end{aligned}
\tag{8}
$$

Additionally, to obtain a smooth model in the inverse problem, a smooth matrix with moving average is designed and embedded into the mutation operation, such as

$$
\mathbf{v}_i^G = \mathbf{m}_i^G + F_i \cdot \Big( \mathbf{m}_{pbest}^G - \mathbf{m}_i^G + \Delta_i \cdot \big[ \mathbf{S}^4 \big( \mathbf{m}_{r1}^G - \widetilde{\mathbf{m}}_{r2}^G \big) \big] \Big),
\tag{9}
$$

which is described in detail in paper [29,37].

In the original DE algorithm, the fitness of a trial vector is directly calculated by vector $\mathbf{u}$. However, this method ignores the fact that the crossover operation can destroy the spatial continuity of a mutant vector. Therefore, in this work, for a trial vector $\mathbf{u}_i$, the fitness is defined as

$$
\begin{aligned}
\Phi(\mathbf{u}_i) &:= \Phi(\mathbf{u}_{s,i}), \\
\mathbf{u}_{s,i} &= \mathbf{S}^4 \mathbf{u}_i
\end{aligned}
\tag{10}
$$

Similarly, the fitness of a target vector $\mathbf{m}_i$ is obtained by the following:

$$\Phi(\mathbf{m}_i) := \Phi(\mathbf{m}_{s,i}),$$
$$\mathbf{m}_{s,i} = \mathbf{S}^2 \mathbf{m}_i \tag{11}$$

*2.3. Magnetic Inversion*

2.3.1. Inversion Method

The inversion optimization process seeks to minimize the discrepancy between observed and predicted data derived from the estimated model. To achieve this, the following objective function is employed:

$$\Phi(\mathbf{m}) = \Phi_d(\mathbf{m}) + \lambda \Phi_m(\mathbf{m}) ,$$
$$\text{s.t. } \mathbf{m}_l \le \mathbf{m} \le \mathbf{m}_u. \tag{12}$$

where $\lambda$ is a regularization factor, $\Phi_d(\mathbf{m})$ denotes the data misfit objective function, $\Phi_m(\mathbf{m})$ represents the model misfit objective function, and vectors $\mathbf{m}_l$ and $\mathbf{m}_u$ are the lower and upper boundary constraints of parameter $\mathbf{m}$, respectively. In this work, a normalized data misfit function is used, which is defined as

$$\Phi_d(\mathbf{m}) = \left\| W_d \left( \boldsymbol{d}^{obs} - \mathcal{F}(\boldsymbol{m}) \right) \right\|_2^2,$$
$$\mathbf{m}_s = \mathbf{S}^2 \mathbf{m},$$
$$W_d = diag \left( \frac{1}{|d_1^{obs}| + \epsilon}, \frac{1}{|d_2^{obs}| + \epsilon}, \cdots, \frac{1}{|d_N^{obs}| + \epsilon} \right),$$
$$\epsilon = std \left( \left| \boldsymbol{d}^{obs} \right| \right), \tag{13}$$

where $W_d$ is a diagonal weighting matrix containing information about the observed data and can be found in [33,38], and $\boldsymbol{d}^{obs} = \left( d_1^{obs}, d_2^{obs}, \cdots, d_N^{obs} \right)$ is the observation data with size $N$. $\epsilon$ represents the standard deviation of vector $\boldsymbol{d}^{obs}$, the purpose of which is to avoid the appearance of 0 in the denominator. In Equation (12), the model misfit $\Phi_m(\mathbf{m})$ is calculated according to the following equation:

$$\Phi_m(\mathbf{m}) = \sum_{i=1}^{M} W_{m,i} |m_i - m_{\text{ref},i}|^1, \tag{14}$$

where $\mathbf{m}_{\text{ref}} = (m_{\text{ref},1}, m_{\text{ref},2}, \cdots, m_{\text{ref},M})$ is the reference model, and $W_{m,i}$ denotes the weight of the $i$th parameter in the model vector $\mathbf{m}$. Generally, $\mathbf{W}_m = (W_{m,1}, W_{m,2}, \cdots, W_{m,M})$ is constructed with the prior information and is calculated by the following [32]:

$$W_{m,i} = \frac{W_{z,i} V_i}{\sum_{k=1}^{M} W_{z,k} V_k},$$
$$W_{z,i} = \frac{1}{D_{z,i}^{\beta}} , \tag{15}$$

where $V_i$ is the volume size of the $i$th element, $W_z$ denotes the depth weight, $D_z$ represents the distance between the observation height and element center, and $\beta$ is the depth factor and is set to 2 for magnetic data inversion [12]. Finally, combining Equations (13) and (14), the objective function of magnetic inversion is summarized as

$$\Phi_m(\mathbf{m}) = \left\| W_d \left( \boldsymbol{d}^{obs} - \mathcal{F}(\boldsymbol{m}) \right) \right\|_2^2 + \lambda \sum_{i=1}^{M} W_{m,i} |m_i - m_{\text{ref},i}|^1, \tag{16}$$

2.3.2. Adaptation of Regularization Factor

The regularization factor plays an important role in the inversion process. In this work, the adaptive method is based on [29] and introduces a strategy to prevent the

value of $\lambda$ from decreasing excessively during the inversion process, which could render the constraints ineffective. To update $\lambda$, the following formula is used. More detailed information can be found in paper [29].

$$\lambda^G = \begin{cases} \max\left(\lambda_{min}, 0.9\cdot\lambda^{G-1}\right) \text{ if } \frac{r_{dm}^G}{r_{dm}^{G-1}} > 1 \\ \lambda^{G-1}, \text{ otherwise} \end{cases},$$

$$r_{dm}^G = \frac{\Phi_{d,\ mean}^G}{\Phi_{m,mean}^G}, \tag{17}$$

## 3. Test and Application

In this section, the proposed algorithm in this paper and JADE are applied to invert both the synthetic data and the synthetic data with noise to validate the effectiveness and robustness of the proposed algorithm. Furthermore, the algorithm is applied to the field data.

### 3.1. Synthetic Data

To validate the correctness of the algorithm, four models are designed as shown in Figure 3. There are 41 points evenly distributed on the ground, and the distance between each point is 10 m. The background magnetic susceptibility is set to 0, and the magnetic susceptibility of the anomalies is set to 0.1 SI. There are 40 cells in the x direction, which is the number of points minus 1, and 29 cells in the z direction, so there are 1160 grids to be inverted.

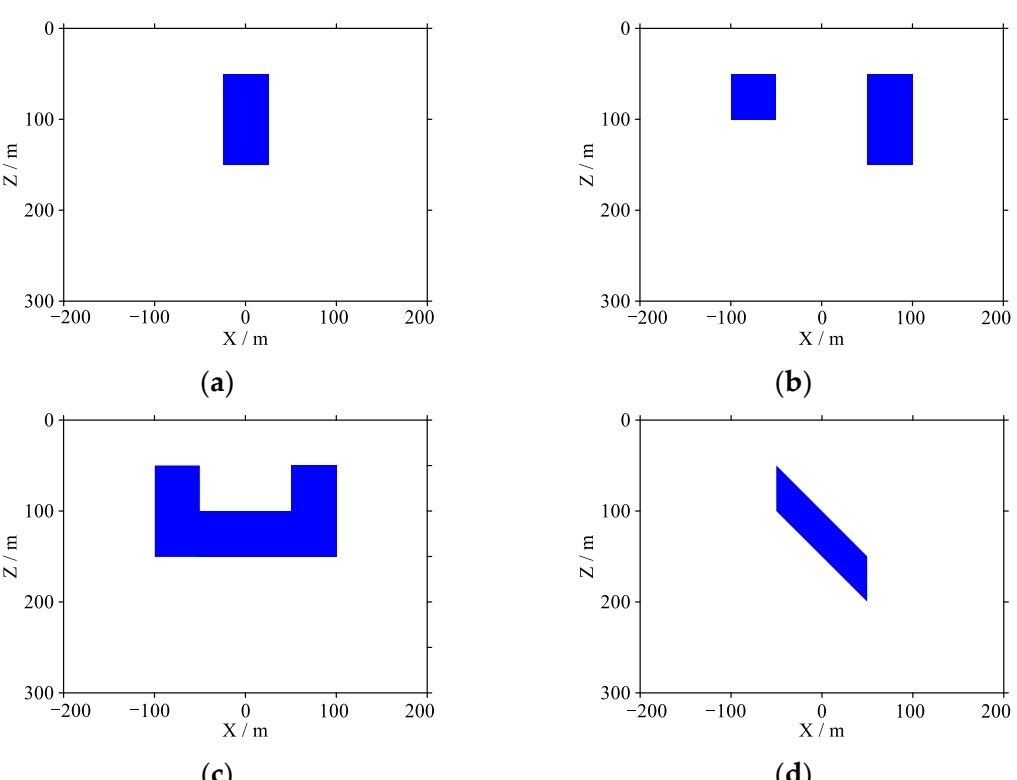

**Figure 3.** Synthetic models for magnetic inversion: (**a**) rectangle; (**b**) parallel rectangle; (**c**) U shape; (**d**) parallelogram.

The synthetic data are obtained by forward modeling, then the inversion is performed using three versions of the algorithms: (1) the JADE algorithms with smooth matrix shown in Equations (10) and (11) and denoted as IADE-1; (2) based on IADE-1, the descent direction is adopted as shown in Equation (9) and denoted as IADE-2; (3) based on IADE-2, the new mechanism for CR generation is added, which is the proposed method in the

previous section, and denoted as IADE. In order to make a fair comparison, the max iterations for the synthetic data are set to 300, both $\mu_F$ and $\mu_{CR}$ are set to 0.5, and the population size for all compared algorithms is 100.

The inversion results of IADE-1 and the anomaly location are shown in Figure 4. It can be observed that the inversion results can generally reflect the locations of anomalies, but they exhibit limited contrast with the background, and some spurious anomalies are introduced at the boundaries. Especially in the parallel rectangle, smaller anomalies are difficult to distinguish in the inversion results, and the anomalies at the boundaries are more obvious.

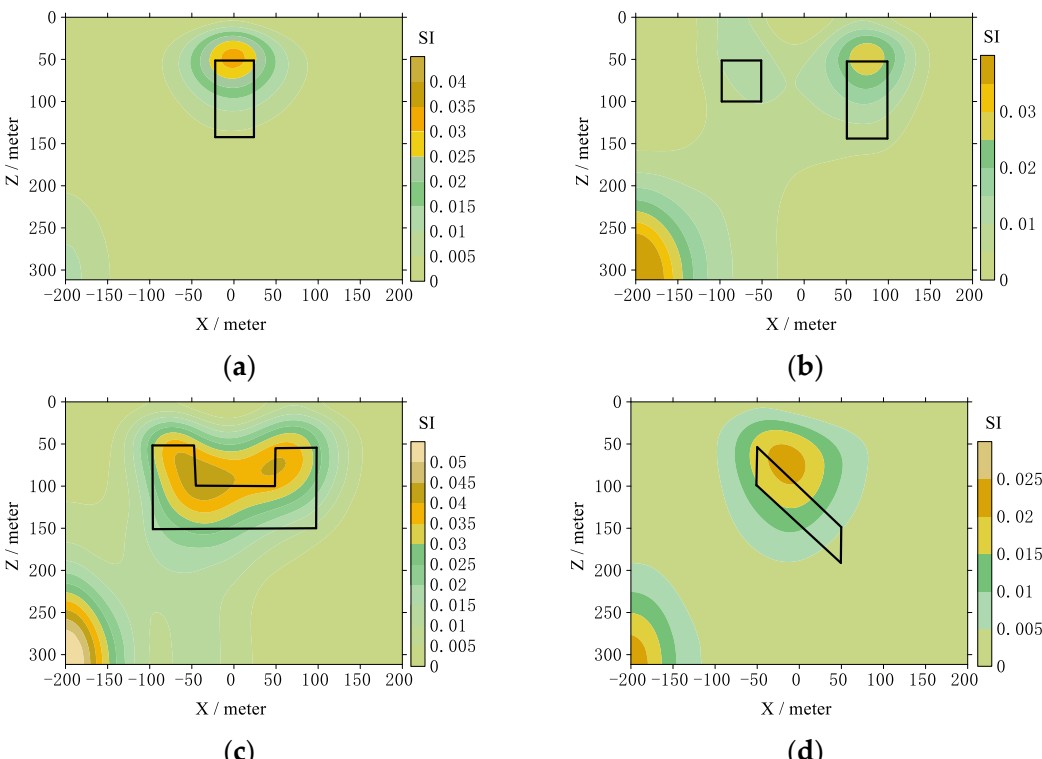

**Figure 4.** The inversion results of synthetic data by IADE-1 for each model: (**a**) rectangle; (**b**) parallel rectangle; (**c**) U shape; (**d**) parallelogram.

The inversion results of IADE-2 and the anomaly locations are shown in Figure 5; also, the results can generally reflect the location of anomalies. Compared with the results of IADE-1, the magnitude of anomalies is larger, and the anomalies at the boundaries are relatively smaller.

The inversion results of IADE and the anomaly location are shown in Figure 6; IADE accurately reflects the position and distribution of the anomaly bodies, with minimal anomalies at the boundaries which closely resemble the background. The inversion results also show significant magnitudes, and in the case of the second model, even the smaller anomaly bodies are well represented in the inversion results.

All methods reveal the magnetic susceptibility distribution of the designed model. In the case of IADE-1 and IADE-2, the anomalies at the boundaries are obvious, while IADE produces comparatively superior results. All three methods perform well in the U-shaped anomaly, which is attributed to its widespread distribution. Among the methods evaluated, IADE yields the best inversion results for the U-shaped anomaly, providing clearer boundaries.

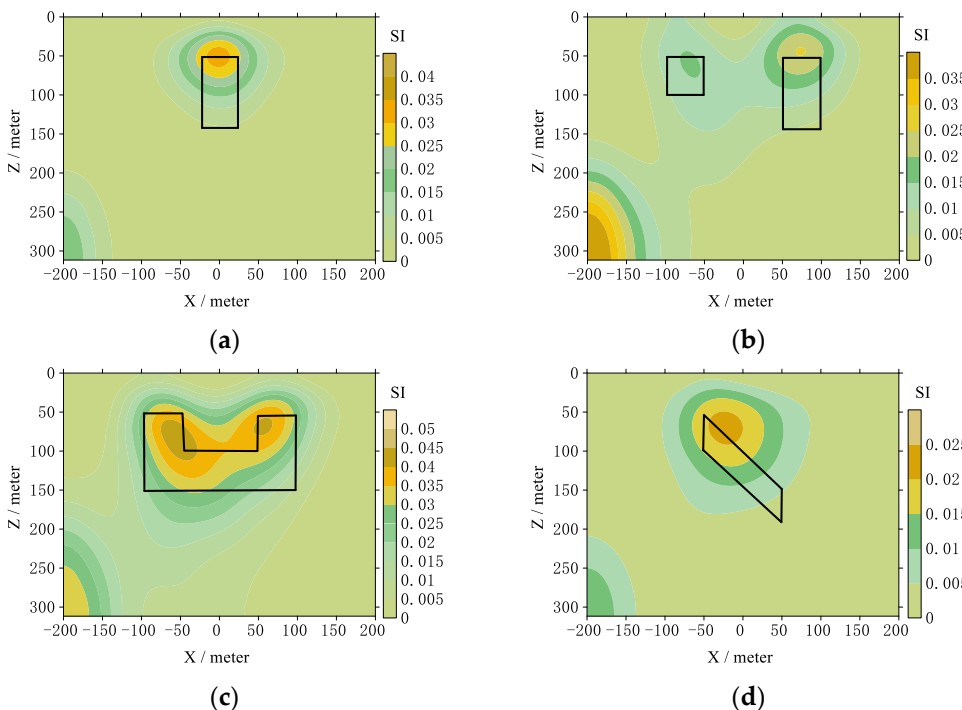

**Figure 5.** The inversion results of synthetic data by IADE-2 for each model: (**a**) rectangle; (**b**) parallel rectangle; (**c**) U shape; (**d**) parallelogram.

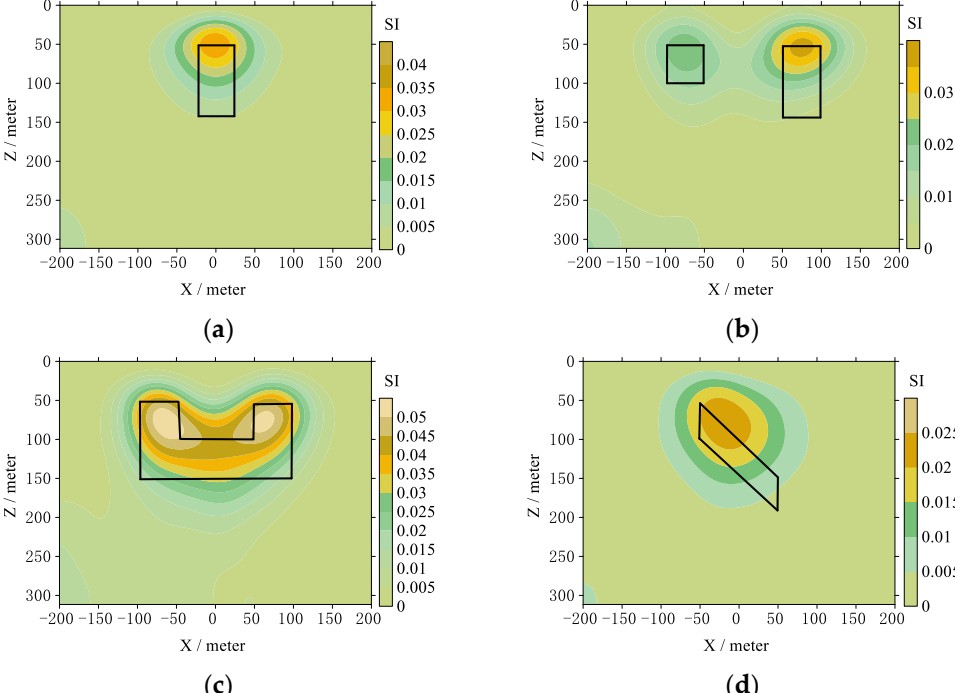

**Figure 6.** The inversion results of synthetic data by IADE for each model: (**a**) rectangle; (**b**) parallel rectangle; (**c**) U shape; (**d**) parallelogram.

Figure 7 shows the convergence curves of the three algorithms for four models during the 300 iterations. It can be seen from the figure that simply incorporating the descent direction in the mutation stage (IADE-2) does not effectively accelerate the convergence of the algorithm. However, the previous inversion results show that the introduction of the descent direction can effectively improve the inversion results. When the descent direction

in the mutation stage and the new mechanism for CR generation are added to the algorithm, the IADE exhibits the characteristics of faster convergence and better results.

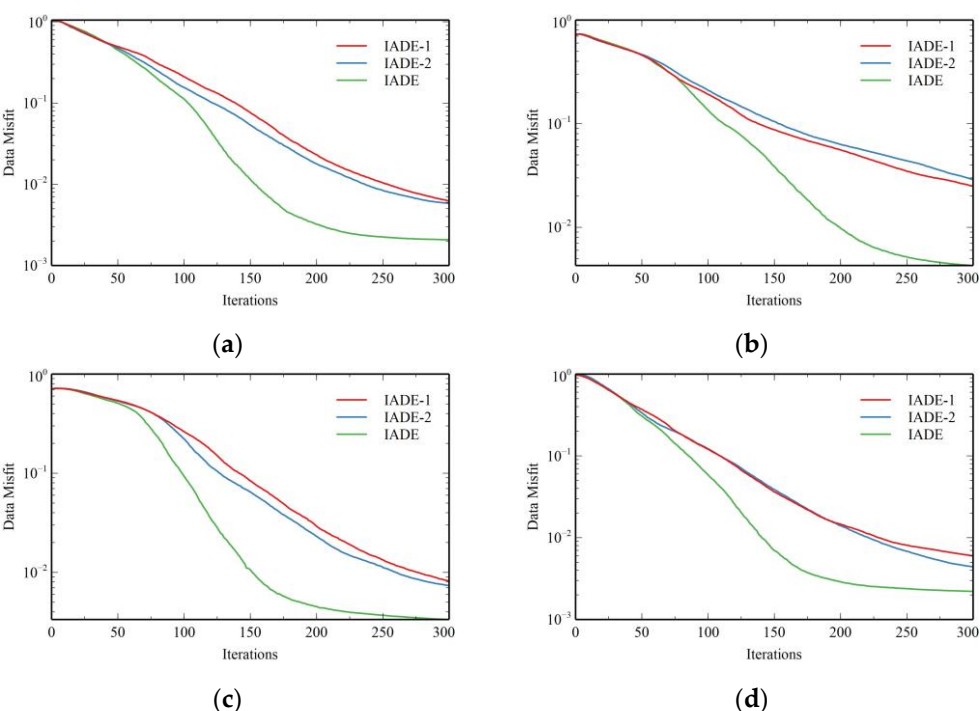

**Figure 7.** The data misfit by different methods for each model: (**a**) rectangle; (**b**) parallel rectangle; (**c**) U shape; (**d**) parallelogram.

### 3.2. Synthetic Data with Noise

To test the robustness of the algorithm, random noise is added to the synthetic data of model d (parallelogram) in Figure 3. Noise levels of 1%, 5%, and 10% are added separately, then the proposed method in this paper is used for inversion. The data with noise are shown in Figure 8, and the inversion results are shown in the figure.

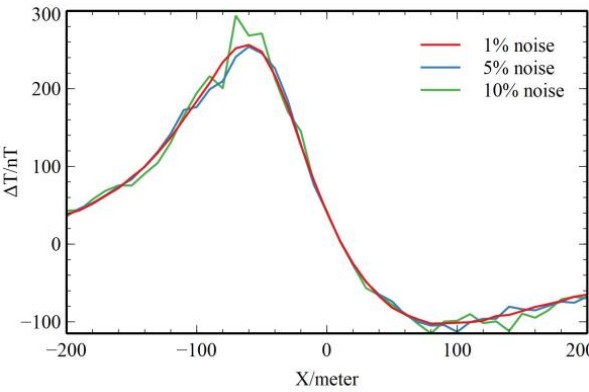

**Figure 8.** Synthetic data of model d (parallelogram) with 1%, 5%, and 10% noise.

From Figure 9, it can be observed that when 1% noise is added, the inversion profile is consistent with the original one. As the noise level increases, spurious anomalies near the boundaries begin to appear, but the position and amplitude of the anomaly are still consistent with the designed model. This indicates that even with certain levels of noise, the algorithm proposed in this paper can still effectively recover the underlying magnetic anomalies. Therefore, the proposed algorithm demonstrates strong robustness.

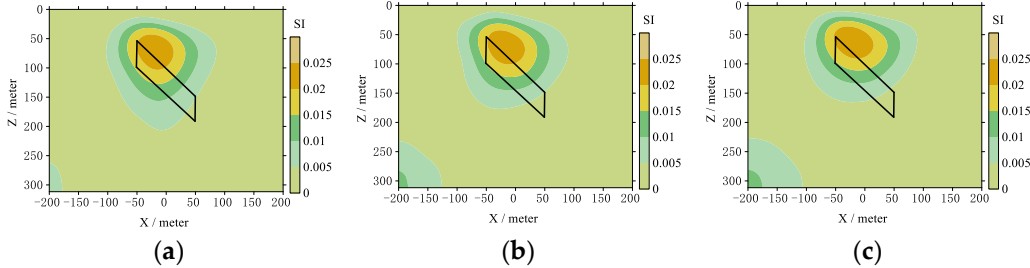

**Figure 9.** The inversion results of synthetic data from IADE for parallelogram model with different noise. (**a**) Noise level: 1%; (**b**) noise level: 5%; (**c**) noise level: 10%.

### 3.3. Field Case: Iron Deposit Prospection of Shihe, Shanxi, China

The Shihe iron ore deposit is located in Shanxi, China. Geotectonically, it is situated northeast of the Hengshan–Wutaishan dome in the Lvliang–Taihang fault block in North China. The primary faults in this region are oriented in the northeast and northwest directions. The study area is covered by Quaternary sediments of the Malan group, with a thickness ranging from 240 to 340 m, as shown in Figure 10. And the magnetic properties of the ore and rock samples are provided in Table 1. It can be found that the magnetite quartzite samples show strong magnetism, while the samples of the other three rocks (plagioclase amphibolite, biotite granulite, and hornblende–plagioclase) have relatively weaker magnetism. This forms a solid foundation for conducting magnetic exploration in this area. Using the observed curve from Figure 11a, it can be inferred that the amplitude anomaly in the observed data is caused by magnetite quartzite, which located near the 600 m mark on the survey line. In the Shihe area, the geomagnetic field $B_0 = 54,463$ nT, the geomagnetic inclination $I_0 = 58°$, and the geomagnetic declination $D_0 = 5°$. The direction of the survey line is 162°, and the length of the survey line is 1880 m. There are 95 magnetic measuring points, and the point space is 20 m. The inversion profile is divided into 3008 cells ($94 \times 32$; there are 94 cells along x direction and 29 cells in z direction).

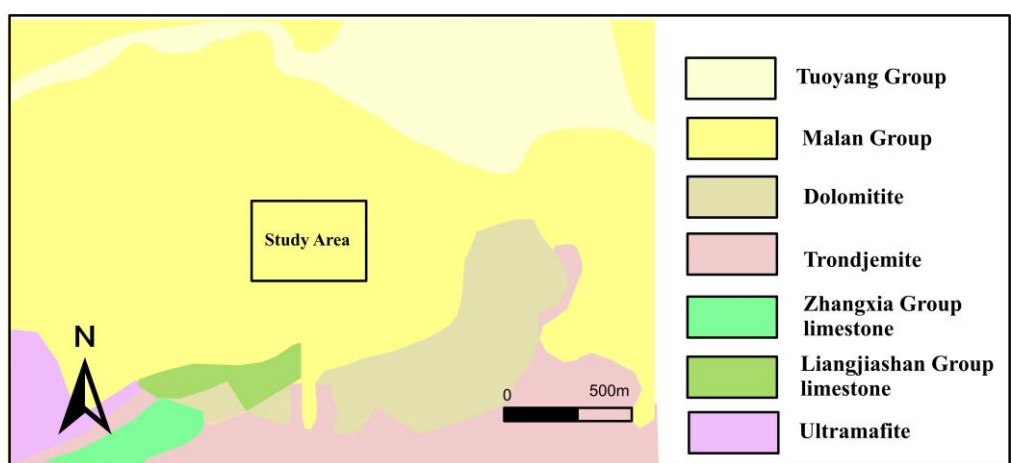

**Figure 10.** The simple geological map of the Shihe deposit in Shanxi, China, from [37].

**Table 1.** Magnetic properties of minerals and rocks in the Shihe area.

| Ores and Rocks | Sample Number | Mean κ ($4\pi \times 10^{-6}$ SI) |
| --- | --- | --- |
| Plagioclase amphibolite | 41 | $4.5 \times 10^2$ |
| Magnetite quartzite | 30 | $1.79 \times 10^5$ |
| Biotite granulite | 27 | $4.27 \times 10^2$ |
| Hornblende–plagioclase | 7 | $3.52 \times 10^2$ |

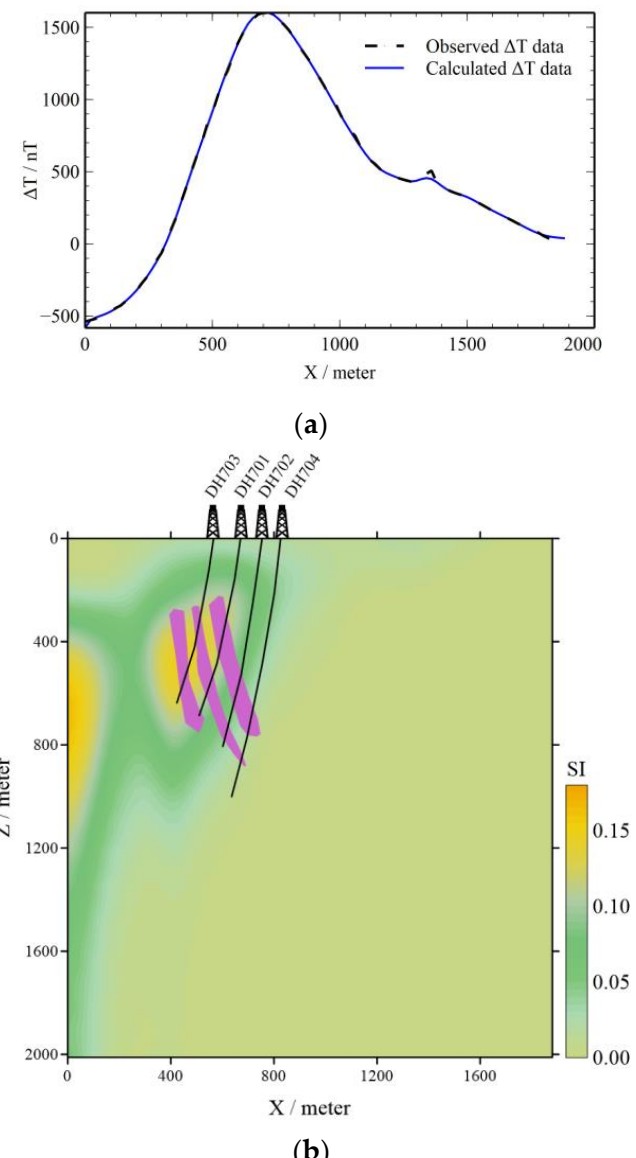

**Figure 11.** The inversion results of magnetic data. The violet polygons denote iron ore bodies, and the black lines denote the drilling wells. (**a**) The observed and calculated data. (**b**) The inverted magnetic susceptibility.

To investigate the ore distribution in the subsurface of the study area, the algorithm proposed in this paper is applied to the magnetic observed data shown in Figure 11a, and the resulting calculated data are depicted in Figure 11a together. The inverted magnetic susceptibility is shown in Figure 11b.

In Figure 11b, the violet polygons in the figure represent the distribution of the ores (magnetite quartzite), which is plotted based on drilling data. Based on the distribution of inverted susceptibility, it is evident that the ore bodies are buried at a depth of approximately 260 m and extend to about 600 m in the z direction. Furthermore, when compared to the borehole data, a significant correlation is observed. However, the terminal portion of ore bodies cannot be accurately depicted by the developed method. This is due to the fact that the magnetic anomaly responses from deep-buried sources are weak and can be easily influenced by noise. As stated by Li and Oldenburg [39], to enhance resolution in both the vertical and deep directions, it is imperative to integrate borehole magnetic data with the ground dataset.

## 4. Conclusions

The main objective of this study is to develop a fast and robust inversion algorithm for magnetic data based on differential evolution. The JADE algorithm is selected and improved according to the analysis of previous studies. To realize our objective, two significant improvements were implemented. First, a new method was introduced to generate the crossover rate to increase the probability of smaller CR values corresponding to better solutions. Second, a new variable $\Delta_i$ was added in the mutation process to keep consistent the perturbation direction that reduces the objective function, thereby improving the local exploitation ability of the algorithm. In addition, the adaptive regularization factor is modified to avoid it becoming too small in the later stage of inversion so as to ensure its constraint effect.

Furthermore, the presented algorithm exhibited significant robustness and stability, as evidenced by the inverted results from both noisy data and field data. Moreover, the inverted model's susceptibility distribution from field data is consistent with that inferred from boreholes. Our work demonstrated the performance of the DE method for magnetic inversion with synthetic and field examples. In our future work, the proposed method will be further enhanced and extended to solve three-dimensional gravity and magnetic inverse problems.

**Author Contributions:** Conceptualization, T.S.; methodology, T.S. and L.C.; software, T.S. and T.X.; formal analysis, T.S., J.H. and B.Z.; resources, L.C.; writing—original draft preparation, T.S.; writing—review and editing, T.S., J.H. and B.Z.; visualization, L.C.; funding acquisition, J.H. and B.Z. All authors have read and agreed to the published version of the manuscript.

**Funding:** This research was funded by the special funding of the Guiyang Science and Technology Bureau and Guiyang University [GYU-KY-[2021]] and the Guizhou Provincial Science and Technology Plan Project (Qian Ke He Jichu-ZK [2021] general type 206).

**Data Availability Statement:** The data of this study are available from the authors upon request.

**Acknowledgments:** The authors wish to acknowledge the Shanxi Institute of Geophysical and Geochemical Exploration for providing magnetic data from the Shihe deposit.

**Conflicts of Interest:** The authors declare no conflict of interest.

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
