# Peer review of "Magnetic Inversion through a Modified Adaptive Differential Evolution"

_minerals, doi:10.3390/min13121518_

Round 1

Reviewer 1 Report

Comments and Suggestions for Authors

This manuscript presents an application of the differential evolution algorithm (more precisely, a modified version of it) to the inversion of magnetic profile data. I recommend to address some major corrections before considering it for publication:

1) There are too many language mistakes in the manuscript. I had to stop correcting them on Page 4, otherwise I wouldn't be able to send the revision on time. The manuscript should be properly revised for grammar and English usage before the next revision.

2) In the paragraph about Differential Evolution from the Introduction, please mention previous works on the inversion of magnetic profile data that use this technique. References [28,32,33] can be used for this purpose, although there are more examples of such use.

3) Section 2.1 lacks information and needs notation corrections:

- Please write all vectors with bold face. Please also change the notation of the forward operator (F(m)), otherwise it will be confounded with the scale factor F.

- As it sounds very unusual that a linear problem becomes nonlinear due to spatial discretization, I believe that the authors meant something else with the sentence on L76-77. In any case, please indicate which physical/geometrical parameters are included the vector m and write down the forward operator F considered in your manuscript. If the operator F involves long formulas, please describe it in words (e.g., "total magnetic anomaly of an infinite [or finite?] prismatic source") and cite the reference that has the formulas used in your code.

4) On Lines 155-156, the definition of the data weighting matrix Wd sounds unusual - usually only the standard deviation is present). Please provide one or more references of other works (at least one reference from another research group) that adopt the definition that you use. Morover, please specify the values of the standard deviation that were selected in each example from Section 3.

5) The tick, axis, and colorbar labels of most figures are too small. Their fonts should have the same size as the fonts in the figure captions.

6) The first paragraph of Section 4 (Conclusions, Lines 266-276) sounds like an introduction paragraph. This section could rather present, for instance, a discussion about the results of Subsection 3.5 - they are interesting, but could have been more commented.  

7) Some detailed corrections (until Page 4)

L19: CR -> crossover rate

L20: make -> tends to reduce (and remove "tend to be smaller")

L23: Please rephrase "some improvements have also been made" to clearly state that the regularization strategy has been previously proposed (in [27], as written in Lines 166-170).

L25: and field -> and a field

L27: For field -> For the field

L50: ES -> EA

L58: Sentence "So, all this ... we face" sounds too informal, please rephrase it

L60: please add a brief description of JADE: "an improved version of JADE, **** [25], and" 

L62: [26], -> [26], by

L63: is capable of -> , we are capable to

L67: Observed from literature [27,28], one can see that a proper-> As observed from the literature [27,28], a proper 

L82: represented -> described

L88: Phi(m) alone is not an optimization problem, it is just a function. Please state the optimization problem correctly, or rephrase "The optimization problem can be expressed"

L92: the new mutation -> new mutation

L93: mechanism -> mechanisms

L95: follow -> follows

L95-96: what does "DE/rand/1" in equation (2) mean?

L99: the inversion work -> our inversion work

L101: what does "DE/current-to-pbest/1" mean?

L106-108: The reviewer could not find in Figure 2 what is claimed in the paragraph. There is no equation for m^G_pbest or m^G_r2 for the 100p% criterion.

L114: please define the Lehmer mean

L124: it’s more -> it is

L124: if reduce -> than to reduce

L124: component -> components

Comments on the Quality of English Language

Please see the previous item

Author Response

Thank you very much for taking the time to review this manuscript. Please find the detailed responses below and the corresponding revisions highlighted track changes in the re-submitted files.

Reviewer 2 Report

Comments and Suggestions for Authors

Your Abstract is too technical and not sufficiently attractive.

Surprisingly, you don’t have any reference from University of British Columbia and Colorado School of Mines such as from Oldenburg and Li which have been pioneers in the development and applications of magnetic inversion methods. This is a link to UBC publication web site: https://gif.eos.ubc.ca/publications/journal.

Line 19: CR not defined … for CR generation is proposed, which

Line 80: EA is not defined.

Line 180: What is the cell size and the number of cells in x, y, and z directions? In summary, the total number of cells.

Figure 3: Your model are presented only in section view (z,y)? What is the representation in plan view? If it is 2D, this is not representative of real magnetic data.

Figure 7 caption: misfit

3.5. There are numerous information that are missing related to your field case: map of the magnetic data, map of the predicted magnetic field, average total field intensity, magnetic inclination, magnetic declination, size of the volume  inverted (cell size, number of cells in x, y, z, total number of cells). Did you have to extract a regional field? If yes, how? Number of iterations. Etc.

Line 268: was chosen and improved.

Comments on the Quality of English Language

I give an example of sentences that should be rewritten and simplified:

To achieve our goal, two signifi- 268

cant enhancements to the algorithm are made. Firstly, introducing a novel approach for 269

generating the crossover rate to increase the likelihood of smaller CR values correspond- 270

ing to better solutions, thereby preserving valuable information from excellent solutions. 271

Secondly, introducing a new variable ?? during the mutation process to keep the pertur- 272

bation direction consistently in a direction that decreases the objective function, thus en- 273

hancing the algorithm's local exploitation capability.

Author Response

(The authors gave the same response as above.)

Round 2

Reviewer 1 Report

Comments and Suggestions for Authors

In general the manuscript has greatly improved after revised by the authors, but some major corrections are still needed in the theory/methodology:

1) The issue of a linear problem becoming nonlinear due to spatial discretization is still unclear in Section 2.1. Reference [36] made this point more clear by commenting (page 81) on the fact that they use the differential equation domain formulation instead of the integral equation domain formulation. It is the formulation, not the discretization, that renders the forward operator nonlinear. Some corrections will be proposed below, in this sense.

2) Regarding the weighting matrix Wd, equation (3) of the new reference (Sharma and Biswas 2013) provides the entire misfit functional, not the specific definition of Wd. Please write down the details of how Wd in equation (13) of the manuscript is obtained from equation (3) of Sharma and Biswas (2013). Otherwise, please replace your entire equation (13) by their (3), adapting their notations for phi, V^0_i, V^c_i, and V^0_max/min. 

3) There are still tiny fonts within the figures. For instance, please compare "X/meter" and "SI" in the upper and lower maps in Figure 5d. Please make sure that all figures have the same (or larger) font size as in the lower Fig. 5d.

Detailed corrections:

L18: Considering that the -> The [[the term "Considering" calls for a subsequent argument that is not provided before the sentence ends]] 

L19: minimizes -> minimize

L20: "two points" sounds more appropriate than "twofold"

L91: magnetization susceptibility -> magnetization susceptibility at each cell of a rectangular grid [[replace "rectangular" with the appropriate type of grid, if needed]]

L91: in the "data derived", please replace "data" by the physical quantity (magnetic anomaly?)

L92: In the conventional magnetic forward and inversion program -> In the integral equation domain formulation

L95-97: a matrix and also ... literature [36] -> linear as we use a differential equation domain formulation [36]

L111: follows -> finding m so that the functional [[please write "m" in bold font]]

L112: Where -> is minimized, where

L112-113: remove ", and the objective ... minimized" 

L133-137: The reviewer could not find in Figure 2 what is claimed in the paragraph. There is no equation for m^G_pbest or m^G_r2 for the 100p% criterion.

L153: it's -> it is 

L153-154: if reduce ... vector -> if the number of components from mutant vector is reduced

L224: by "abnormality" did you mean "source"? 

Comments on the Quality of English Language

English usage has significantly improved in the revised version

Author Response

1) The issue of a linear problem becoming nonlinear due to spatial discretization is still unclear in Section 2.1. Reference [36] made this point more clear by commenting (page 81) on the fact that they use the differential equation domain formulation instead of the integral equation domain formulation. It is the formulation, not the discretization, that renders the forward operator nonlinear. Some corrections will be proposed below, in this sense.

Responses: We have made changes as you suggested.

2) Regarding the weighting matrix Wd, equation (3) of the new reference (Sharma and Biswas 2013) provides the entire misfit functional, not the specific definition of Wd. Please write down the details of how Wd in equation (13) of the manuscript is obtained from equation (3) of Sharma and Biswas (2013). Otherwise, please replace your entire equation (13) by their (3), adapting their notations for phi, V^0_i, V^c_i, and V^0_max/min.

Responses: We make some changes in equation (13). Now our  is almost the same as reference (Lelièvre, P.G.; Oldenburg 2006) equation (17). Equation (17) in (Lelièvre, P.G.; Oldenburg 2006) and equation (3) in (Sharma and Biswas 2013) are different in form, but similar in outcome. Equation (3) in (Sharma and Biswas 2013) add  in the denominator to avoid dividing by a very small number(it is a constant), and in our implements, the STD is adopted, also it is a constant. We've removed potentially confusing parts in equation (13), that is  in denominator, the purpose of this part is similar to 1/N in equation (3) [Sharma and Biswas 2013], and removing it has no effect on the results.

3) There are still tiny fonts within the figures. For instance, please compare "X/meter" and "SI" in the upper and lower maps in Figure 5d. Please make sure that all figures have the same (or larger) font size as in the lower Fig. 5d.

Responses: We changed the font to the same size now.

Detailed corrections:

Responses: Thank you for your revision of our manuscript. We have made the changes in order according to the suggestions below.

L18: Considering that the -> The [[the term "Considering" calls for a subsequent argument that is not provided before the sentence ends]]

L19: minimizes -> minimize

L20: "two points" sounds more appropriate than "twofold"

L91: magnetization susceptibility -> magnetization susceptibility at each cell of a rectangular grid [[replace "rectangular" with the appropriate type of grid, if needed]]

L91: in the "data derived", please replace "data" by the physical quantity (magnetic anomaly?)

Responses: We changed it to magnetic data, and can be found in equation (1)[ Liu, S.; Liang, 2018].

L92: In the conventional magnetic forward and inversion program -> In the integral equation domain formulation

L95-97: a matrix and also ... literature [36] -> linear as we use a differential equation domain formulation [36]

L111: follows -> finding m so that the functional [[please write "m" in bold font]]

L112: Where -> is minimized, where

L112-113: remove ", and the objective ... minimized"

L133-137: The reviewer could not find in Figure 2 what is claimed in the paragraph. There is no equation for m^G_pbest or m^G_r2 for the 100p% criterion.

Responses: We add the exact definition of m^G_pbest and m^G_r2 in Figure 2 and it also can be found in table 1 [28](JADE: Adaptive differential evolution with optional external archive).

L153: it's -> it is

L153-154: if reduce ... vector -> if the number of components from mutant vector is reduced

L224: by "abnormality" did you mean "source"?

Responses: Yes, we mean source, and we changed it to "All methods reveal the magnetic susceptibility distribution of the designed model."

Reviewer 2 Report

Comments and Suggestions for Authors

I think you paper is could be accepted if you improved your presentation and English. There several locations where the English is rocky. Furthermore, your techniques are relatively new for the community and it is important to explain them well and avoid confusion. Finally, you should state clearly that your technique has only be applied to 2D data. You should discuss the potential for 3D, or the challenges, since 3D magnetic inversions are the mostly used techniques today as the earth is rarely 2D.

Comments on the Quality of English Language

English is rocky at several locations.

Author Response

I think you paper is could be accepted if you improved your presentation and English. There several locations where the English is rocky. Furthermore, your techniques are relatively new for the community and it is important to explain them well and avoid confusion. Finally, you should state clearly that your technique has only be applied to 2D data. You should discuss the potential for 3D, or the challenges, since 3D magnetic inversions are the mostly used techniques today as the earth is rarely 2D.

Responses: We modified the English expressions in some places.

We modified the JADE flow chart and then modified equation(13) to avoid confusion.

From a methodological point of view, it is feasible to apply the method in this article to three-dimensional magnetic data inversion. However, the current DE algorithm still consumes several hundred thousand forward calculations when running a 2D inversion problem. Therefore, to solve the 3D problems, the rate of convergence of the algorithm requires further enhancement. And we will carry out relevant work later.
